# Electropolishing—A Practical Method for Accessing Voids in Metal Films for Analyses

Sebastian Moser [1,*], Manuel Kleinbichler [1], Sabine Kubicek [1,†], Johannes Zechner [1] and Megan J. Cordill [2,3]

1   KAI Kompetenzzentrum Automobil-und Industrieelektronik GmbH, Europastrasse 8, 9524 Villach, Austria; manuel.kleinbichler@k-ai.at (M.K.); sabine.kubicek@tuwien.ac.at (S.K.); johannes.zechner@k-ai.at (J.Z.)
2   Erich Schmid Institute of Materials Science, Austrian Academy of Sciences, Jahnstrasse 12, 8700 Leoben, Austria; megan.cordill@oeaw.ac.at
3   Department Materials Science, Montanuniversität Leoben, Jahnstrasse 12, 8700 Leoben, Austria
*   Correspondence: sebastian.moser@k-ai.at
†   Present address: Research Area Biochemical Engineering, Institute of Chemical, Environmental and Bioscience Engineering, TU Wien, Gumpendorfer Strasse 1a, 1060 Vienna, Austria.

**Featured Application: Void quantification in the metallization layer of microelectronic chips.**

**Abstract:** In many applications, voids in metals are observed as early degradation features caused by fatigue. In this publication, electropolishing is presented in the context of a novel sample preparation method that is capable of accessing voids in the interior of metal thin films along their lateral direction by material removal. When performed at optimized process parameters, material removal can be well controlled and the surface becomes smooth at the micro scale, resulting in the voids being well distinguishable from the background in scanning electron microscopy images. Compared to conventional cross-sectional sample preparation (embedded mechanical cross-section or focused ion beam), the accessed surface is not constrained by the thickness of the investigated film and laterally resolved void analyses are possible. For demonstrational purposes of this method, the distribution of degradation voids along the metallization of thermo-mechanically stressed microelectronic chips has been quantified.

**Keywords:** electropolishing; void analysis; thin metal film; copper; thermo-mechanical fatigue; scanning electron microscopy; poly-heater

## 1. Introduction

Voids in copper have been reported in mechanical tensile and impact testing [1–3], mechanical and electrical fatigue testing [4–6], as well as in connection with exposure to irradiation [7,8] and processing related issues [9–11]. In the context of mechanical testing, voids can be considered as very early degradation features that, upon increased loading, grow in size, coalesce, and facilitate ductile crack growth [1,12–14]. Therefore, void analyses have a wide range of applications and, depending on the investigated specimen, a comparison of different possible preparation techniques can be very beneficial.

Traditionally, the most widespread method to analyze voids, particularly suitable for bulk samples, is a metallographic cross-section [15–17] prepared by mechanical grinding and subsequent polishing steps. During these processing steps, plastic deformation of the subsurface [18,19] takes place that potentially results in void smearing [20], and additional chemical preparation might be necessary. For investigation of such prepared samples, typically an optical microscope or a scanning electron microscope (SEM) is used [3,21,22].

When considering thin film specimens, especially structured ones, as in microelectronics applications, focused ion beam (FIB) is a widely used preparation technique for investigating general defects, including voids [23–26]. Compared to embedded mechanical cross-sections, FIB has the advantage that no time-consuming pre-preparation steps are

required and that imaging is also possible between different milling steps, allowing for tomographic void analyses [27]. When it comes down to preparing samples on a larger length scale, which usually is required for statistically sound void analyses, FIB becomes unattractive due to time and cost reasons.

The scope of this publication is to present electropolishing in the novel context of a sample preparation technique for void analyses in metal thin film samples. Generally, electropolishing is an electrochemical process that removes material of an anodically poled metallic work piece and in doing so polishes and deburrs the surface [28–31]. Already in the 20th century, many publications released contributed to obtaining a fundamental understanding of its underlying mechanisms [29,30,32–39]. As of today, recent review publications [40,41] demonstrate that still a lot of research on electropolishing is performed, especially towards the direction of fine tuning process parameters [42] and the chemical composition of the used electrolyte [43,44]; often in the application of specialized industrial manufacturing processes [45–50]. Within the present publication, a simple experimental setup for electropolishing single microelectronic chips with a copper metallization is presented and optimized in terms of micro smoothing results. As a result from the electropolishing process, sub-micrometer-sized degradation features (voids, cracks) in the chips' metallization are not only accessed but can also be well distinguished from the background in electron micrographs, allowing quantitative void analyses. Finally, the major advantage of electropolishing, accessing voids in the lateral direction of the thin film (compared to mechanical cross-section and FIB, accessing the cross-sectional direction or through thickness), is demonstrated by means of a statistically sound, laterally resolved void study.

## 2. Device under Test

The samples used within this work are microelectronic test chips that are referred to as poly-heaters. A detailed description of their inner architecture and of the experimental setup used for actuating them can be found elsewhere [51]. Poly-heaters are provided with integrated active heating (utilizing an electrically resistive, polycrystalline silicon layer) and are specifically designed for studying thermo-mechanical fatigue of metallizations. Primarily, the chips consist of a 120-µm-thick silicon substrate and a 20-µm-thick copper metallization layer. Upon heating, the poly-heater's Cu metallization layer gets significantly strained due to the mismatch in the coefficients of thermal expansion (CTEs) of Si (2.56 ppm/K) and Cu (17 ppm/K). This leads to a corresponding mechanical stress that drives microstructural degradation processes, such as void and crack formation. Figure 1a shows a light microscopy image of a poly-heater. Depending on the purpose of their function one can distinguish between two different types of Cu plate structures: First, there is the structure of interest in terms of thermo-mechanical degradation, which is the 700 µm × 490 µm large Cu plate in the center of the chip. It is located directly above the actively heated region that is available in two different layouts, resulting in different temperature distributions in the Cu, as will be shown in Section 4. Secondly, there are eighteen outer Cu plates that act as electrical contact pads for heating (the two larger rectangular pads of size 1300 µm × 280 µm) and for performing measurements (the remaining sixteen smaller pads). For example, some of these measurement pads are connected to the central Cu plate (see Figure 1a) and allow for a measurement of its electrical resistance, serving as a parameter quantifying its degradation.

For later degradation void analyses (Section 4) two poly-heater specimens have undergone 1000 heat cycles between 100 °C and 400 °C, with a heating duration of 200 µs and a repetition frequency of 1 Hz. These temperatures refer to the ones measured using the poly-heaters' integrated temperature sensor [51], which is located below the central Cu plate at a distance of 245 µm from its edge.

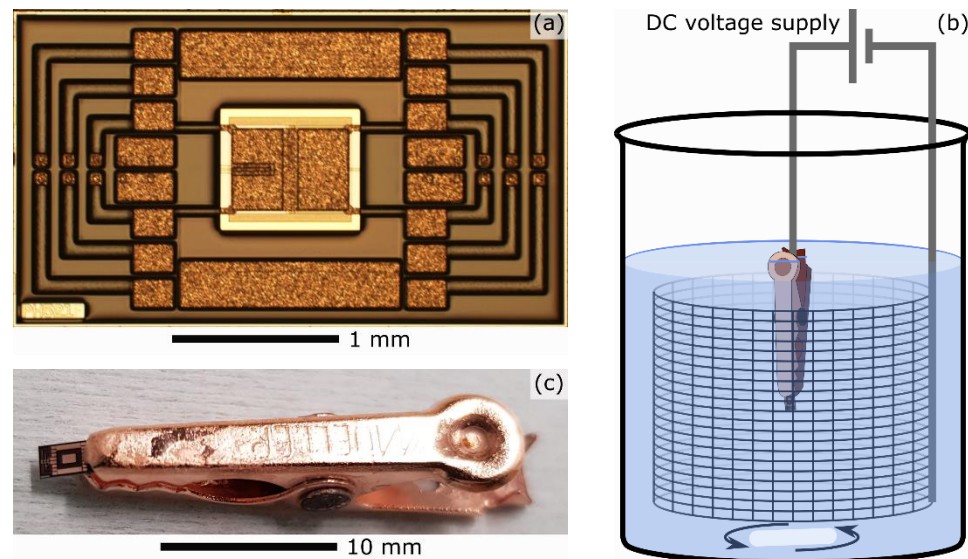

**Figure 1.** (**a**) Light microscopy image of a poly-heater. (**b**) Schematic diagram of the electropolishing setup and (**c**) an alligator clip electrically contacting a poly-heater for electropolishing.

## 3. Electropolishing Setup

A schematic representation of the setup used for electropolishing (a 2-electrode setup) is shown in Figure 1b. This rather simple setup comprises a DC voltage supply, a beaker, in which the two electrodes are in electrical contact with each other via an electrolyte, and a magnetic stirring system. The specific electrolyte used is well-suited for electropolishing copper and is composed of 45 vol% orthophosphoric acid (concentration 85%), 45 vol% methanol, and 10 vol% isopropanol. For the electropolishing experiments presented in this work the poly-heater specimens were electrically contacted by an alligator clip, as shown in Figure 1c. To allow electropolishing of the central Cu plate without damaging it, the alligator clip was used to contact one of the measurement pads that are internally connected to the central plate. By using an alligator clip made of copper, no additional metal was introduced to the system, minimizing measurement errors due to the formation of a galvanic coupling and possible galvanic corrosion of the chip's electropolished surface [52,53]. As depicted in Figure 1b, the alligator clip clamping the poly-heater to be electropolished is fully immersed into the electrolyte and acts as an anode. By this means, the contact surface between the anode and the electrolyte, determined by the electrically contacted Cu surface of the poly-heater ($\approx$2 mm$^2$) and the alligator clip ($\approx$400 mm$^2$), becomes large enough such that the resulting electrical currents can be measured, and reproducibility of the operating conditions can be guaranteed. A cylindrically shaped mesh grid made of stainless steel with height 42 mm and diameter 16 mm functions as a cathode. All experiments were performed at room temperature and with stirring of a frequency of 200 revolutions per minute.

In addition to the electrode materials and the electrolyte, multiple parameters influence the electropolishing process. These comprise electrode distance, surface area ratio of the cathode with respect to the anode but also applied voltage (and corresponding current density), electropolishing duration, stirring, temperature, electrolyte bath age, etc. [33,38,39,44,54,55]. Yang et al. [40] recommend to acquire a current-voltage curve to find the optimal electropolishing voltage for the particular setup and operating conditions in use. Such a curve characterizing the setup in the present publication is shown in Figure 2. Well in agreement with reports in literature [29,32,44,46,56,57], three different regimes can be identified: the first regime ranges from 0.0 V to roughly 0.7 V and is determined by an increasing current. In this regime, etching of the metal (preferentially at grain boundaries) occurs and for this reason no surface smoothing takes place [30,40]. The second regime (extending from 0.7 V to 2.5 V) is characterized by a constant current density

and denotes the interval, in which the optimal voltage in terms of surface smoothing can usually be found [49,56–58]. In this regime, a viscous liquid layer is formed on the anode's surface, where the continuously dissolving Cu is present at high concentrations in form of aqueous cupric salts. Their transport away from the anode's surface is limited by diffusion processes that manifest in the constant current [29,33,48]. The principle leading to surface smoothening can be explained, as follows. Projections of the anode's surface protrude further into the viscous liquid layer than crevices and for this reason they reach out to regions with lower ionic concentrations. This causes a more effective dissolution and faster material removal of the projections, leading to a gradual levelling of the surface. The third regime corresponds to voltages of above 2.5 V, at which oxygen bubbles [34,46,50] are formed at the anode's surface disturbing the viscous layer. Bubbles that are trapped on the surface potentially result in the formation of severe etch pits [37,47,59].

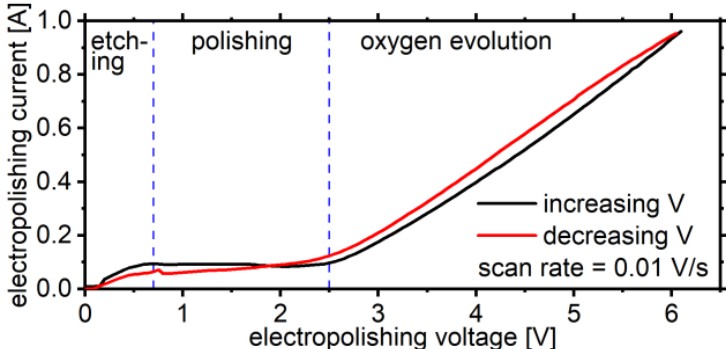

**Figure 2.** Current-voltage curve of the sample-setup configuration used within this work. Optimal electropolishing results in terms of surface smoothing are achieved in the polishing regime, characterized by a constant current density.

An experimental series is presented, where the optimal working point in the current-voltage curve (Figure 2) with respect to surface smoothing has been determined. Within this series pristine (thermo-mechanically non-stressed) poly-heater samples have been electropolished for 360 s at different voltages in the range from 0.0 V to 4.5 V and afterwards their surface roughness has been characterized by confocal laser scanning microscopy (utilizing a Sensofar S neox). Corresponding line profiles (with a projected length $L_0 = 175$ μm and a spacing between data points $\Delta x \approx 0.129$ μm) are depicted in Figure 3a and the profile of the non-electropolished sample is discussed first. In a simplified approach the profile can be interpreted as being composed of a larger-range corrugation that is undulated by short-range roughness features. These short-range features are comprised of micro peaks and micro valleys that can be attributed to grain boundary grooves, as the grains of the investigated Cu are characterized by an equivalent circular diameter of $(2.7 \pm 0.6)$ μm [60,61]. The line profiles of the electropolished samples in Figure 3a qualitatively corroborate the general surface smoothing observations of electropolishing in the different voltage regimes. In particular, one can see that for voltages exceeding 1.0 V micro features are smoothed very effectively. For a quantitative evaluation, the two roughness parameters $R_q$ and the less known tortuosity [62], also referred to as relative line length [63], have been determined from thirteen independent line profiles and are presented as error bar plots in Figure 3b,c. The tortuosity has been calculated according to

$$tortuosity = \frac{L - L_0}{L_0},\qquad(1)$$

where $L_0 = 175$ μm is the before-mentioned projected length of the profile (see Figure 3a) and $L$ is the line integral of the profile. For both $R_q$ and tortuosity (Figure 3b,c) it can be seen that electropolishing at voltages of 0.0 V and 0.5 V clearly does not contribute to surface smoothing. By means of the $R_q$ parameter, one can argue that very good polishing

results can be obtained for voltages between 1.0 V and 2.0 V and that for higher voltages the smoothing becomes less effective. The particular effect of micro levelling [64] grain boundary grooves is very well explained by means of the tortuosity. A highly significant difference between the etched samples (0.0 V and 0.5 V) and the polished samples (>0.5 V) is observed. Especially for the samples polished in the oxygen evolution regime ($\geq$2.5 V), it can be seen that micro levelling is still very effective (Figure 3a,c) and that the slight increase in tortuosity in this regime (inset image in Figure 3c) is attributed to pitting. In agreement with the $R_q$ evaluations, a global minimum of the tortuosity at 2.0 V can be found, denoting the optimal electropolishing voltage.

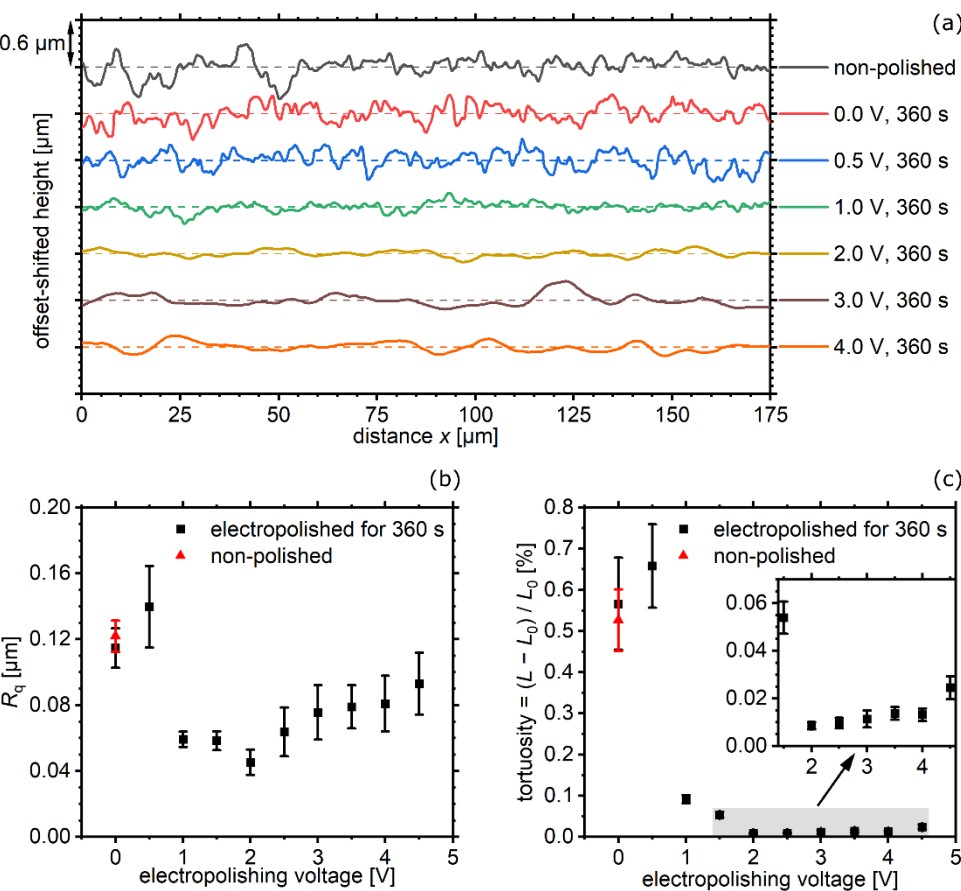

**Figure 3.** (**a**) Representative confocal laser scanning microscopy line profiles from samples electropolished at different voltages as well as the evaluations of (**b**) $R_q$ and (**c**) tortuosity using the average of thirteen line profiles.

The two thermo-mechanically cycled poly-heaters have been electropolished at these optimized parameters (2.0 V for 360 s), resulting in a material removal of $\approx$5 µm, quantified by confocal microscopy. By this means degradation voids in the interior of their Cu metallization have been accessed at a depth corresponding to the material removal, as shown in the cross-sectional schematic illustration in Figure 4a. Subsequent imaging in top view therefore allows quantification of the accessed voids along the lateral direction of the metallization. SEM images of the electropolished surface have been acquired with a JEOL JSM-IT100 InTouchScope™, using an acceleration voltage of 10 kV and a backscatter electron (BSE) detector. A representative image in Figure 4b shows that the depicted voids can be very well distinguished from the background. This is a result from the effective levelling of topographical micro features (e.g., grain boundary grooves) during the electropolishing and allows minimizing systematic errors in subsequent void evaluations by means of digital image processing.

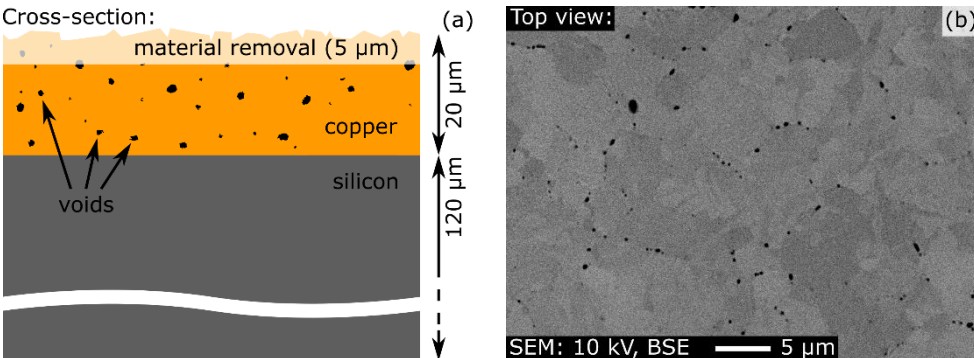

**Figure 4.** (**a**) Cross-sectional schematic illustration visualizing the procedure of accessing voids in the interior of a Cu metallization by material removal during electropolishing. (**b**) Top view SEM image of a thermo-mechanically stressed Cu metallization that has been prepared by electropolishing (2.0 V for 360 s, ≈5 µm material removal) for void analysis.

## 4. Results and Discussion

In the following, a laterally resolved degradation void study is presented, and based on it, the advantages of electropolishing in the preparation of thin films are demonstrated. For the present study, two poly-heaters of different designs are used (Section 2). They are identical in copper, but differ in the functional layers, amongst others, in the polycrystalline silicon that is utilized for heating. During a pulse the central Cu plate of the poly-heater of the first design is homogeneously heated, whereas the one of second design is inhomogeneously heated along its length, being cooler at the center than at the edges. Respective temperature distributions in the Cu at the end of a heat pulse are visualized by the 2-dimensional finite element simulations [65] in Figure 5a,b.

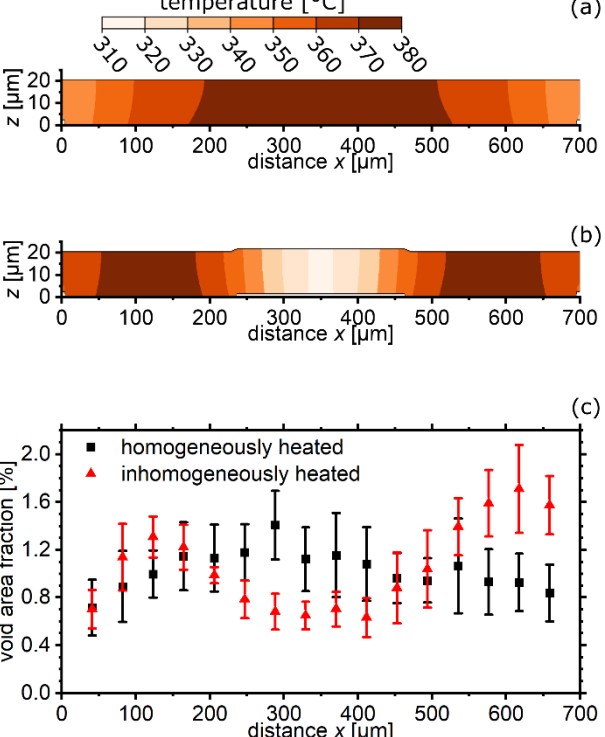

**Figure 5.** (**a**) Cross-sectional temperature distribution in the Cu layer of (**a**) a homogeneously heated and (**b**) an inhomogeneously heated poly-heater, simulated by finite element method [65]. (**c**) Profile of void area fraction along the lateral direction of the investigated poly-heaters' Cu plate.

After thermo-mechanical stress testing and electropolishing (Sections 2 and 3) the 700 μm × 490 μm large central Cu plate (Figure 1a) of the two poly-heaters was imaged by a grid of 16 × 6 SEM images. These micrographs had the same magnification as the representative one shown in Figure 4b. Subsequently, the voids have been extracted using a digital image processing routine [66]. To prevent pixel noise in the SEM images from being erroneously attributed to as voids, a size threshold has been used, by which only objects with an equivalent circular diameter larger than 60 nm are considered in the evaluation. For the individual images the void area fraction, functioning as a parameter for early microstructural degradation, has been computed by dividing the void-represented area by the total imaged area and saved according to the grid of images. The column-wise statistically evaluated data, corresponding to the laterally resolved profile of the void area fraction across the 700-μm-long Cu plate, is presented in Figure 5c. One can see that the homogeneously and the inhomogeneously heated poly-heater are characterized by a different distribution of voids. Regions with higher temperature show more pronounced degradation in terms of void area fraction. The slight asymmetry in the data in Figure 5c is very likely to be caused by an asymmetry in the functional layers of the chip. An additional buffer layer (encasing the temperature sensor) extends over an area of 250 μm × 34 μm below the left half of the Cu plate. This localized layer was not taken into account in the finite element simulations in Figure 5a,b and would lead to slightly decreased temperatures in the left half of the plate.

When discussing about electropolishing as a preparation technique for void analyses in thin films, its major advantage compared to cross-sectional preparation is very clear. A large area for statistically relevant void analyses can be accessed that is not constrained by the film thickness. In case one would want to make a similar study as the one presented in this section on mechanically cross-sectioned specimens and achieve the same statistics, the required effort would be substantially higher. Multiple iterations consisting of grinding, polishing, and SEM imaging would be necessary in order to access a total area that is as large as the one of the electropolished specimen. Put into numbers, for analyzing the voids in the central Cu plate of the poly-heater $\approx$25 iterations of cross-sectioning, each accessing an area of 700 μm × 20 μm = 14,000 μm$^2$, would be necessary to match the area accessed by electropolishing (700 μm × 490 μm = 343,000 μm$^2$). Further advantages of electropolishing are its cost efficiency, ease of use, and that no plastic deformation is introduced to the accessed subsurface, unlike mechanical preparation. This behavior could motivate performing more sophisticated void analyses that are extended by results obtained by electron backscatter diffraction (EBSD), such as kernel average misorientation (KAM). In doing so preferential nucleation sites of voids in the microstructure could be identified.

## 5. Conclusions

It has been shown that electropolishing is a convenient method for preparing metal thin films for void analyses. During the electropolishing process, material is removed from the sample while, at the same time, micro corrugation is levelled out. The accessed surface is very well-suited for interior degradation void analyses. As the voids are accessed in lateral direction of the film, statistical analyses can be performed over a large area. With a conventional cross-sectional preparation (embedded-mechanical cross-section or FIB), this is only possible with a substantially higher effort, as the accessed area is constrained by the low thickness. Electropolishing is not limited to copper; moreover, other metals such as aluminum, silver, gold, titanium, etc., can be prepared after adjustment of the electrolyte and the polishing parameters. Possible applications in terms of preparation for void analyses comprise reliability testing in microelectronics, microelectromechanical systems (MEMS), and flexible electronics but also process control in metal depositions.

**Author Contributions:** Conceptualization, S.M.; methodology, S.M.; software, M.K.; validation, S.M. and M.K.; formal analysis, S.M. and M.K.; investigation, S.M. and S.K.; data curation, S.M. and M.K.; writing—original draft preparation, S.M.; writing—review and editing, S.K., J.Z. and M.J.C.; visualization, S.M.; supervision, S.K. and J.Z.; project administration, J.Z. and M.J.C. All authors have read and agreed to the published version of the manuscript.

**Funding:** This work was funded by the Austrian Research Promotion Agency (FFG, Project No. 881110).

**Institutional Review Board Statement:** Not applicable.

**Informed Consent Statement:** Not applicable.

**Data Availability Statement:** The data presented in this study may be available upon reasonable request by contacting the corresponding author. Restrictions apply to the availability of these data.

**Acknowledgments:** The authors would like to express their gratitude to J. Karlovska for the supervision during the electrochemical experiments and to P. Hoffmann for providing the finite element simulations presented in this publication.

**Conflicts of Interest:** The authors declare no conflict of interest.

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
