# Peer review of "Electropolishing—A Practical Method for Accessing Voids in Metal Films for Analyses"

_applsci, doi:10.3390/app11157009_

Round 1

Reviewer 1 Report

The article presents description of a method to detect internal voids in metals through electropolishing. The subject is interesting and novel, and the paper is well written. The results are presented on a comprehensive and sufficient way, and the overall structure of the paper help the reader to follow the content of the work. A minor remark is this:

I would like the authors to add more information regarding the mechanism that causes metal polishing and how does this depend on the applied voltage. Also, what changes by increasing the voltage that causes degradation of the layer on a different way?

I would suggets the acceptance of the paper for publication, provided that the aforementioned information have been added in the manuscript.

Author Response

We thank the reviewer for this suggestion. A more detailed description of the mechanism of the electropolishing process and the role of the voltage has been added to lines 136-146.

Reviewer 2 Report

Very good work!

Authors can compare other methods and comment on the benefits of their method.

Author Response

We thank the reviewer for this comment. The electropolishing method presented is compared to cross-sections at Lines 239-251. We have revised this paragraph to make the comparisons more clear to the reader. Additionally, a comparison to metallographic and FIB cross-sections was presented in the introduction (Lines 37-51).